

# Co-incidence Analysis of Changes in Flood Magnitude and Shifts in Flood Timing in a Large Tropical Pluvial River Basin

Poulomi Ganguli[1], Yamini Rama Nandamuri[1], Chandranath Chatterjee[1]

[1]Department of Agricultural and Food Engineering, Indian Institute of Technology Kharagpur, West Bengal 721 302, India

*Correspondence to*: Poulomi Ganguli (pganguli@agfe.iitkgp.ac.in; poulomizca@gmail.com)

**Abstract.** Understanding trends in flood severity and the persistence in peak discharge timing along a vast river network is vital for basin-scale flood risk management and reinsurance purposes. While earlier studies have primarily focused on analysis of either trends in floods or its seasonality independently, here for the first time, we assess coincidence of changes in peak discharge and shifts in its timing in one of the

largest peninsular rivers (drainage area of 14 1589 km$^2$), Mahanadi River Basin (MRB), in India during 1970 – 2016. Our research is motivated by the recent six major consecutive floods over MRB during the years 2001, 2003, 2006, 2008, 2011, and 2013. We analyze flood properties using peak fluvial discharge indicators, Monsoonal (from June 1 – end of September, during the Indian summer monsoon period), Maxima Flood (MMF) and Peak over Threshold Flood (POTF) events. While we find a blend of

(insignificant) up/downward trends in flood magnitude at Upper MRB (Region I), the middle reaches of the basin (Region II) showed an upward trend in flood magnitude with a larger number of sites detect significant trends in floods for the POTF events. Although the average dates of peak discharge in the basin are concentrated in August, notwithstanding the nature of flood samplings, a delayed (or earlier) shift in flood timing is apparent for most of sites. Further, we detect potential hotspots, where

up/downward trends in flood magnitude coincide with early (or delayed) dates of flood occurrences. Based on observational evidence, here we show that up to one-third of sites show an up/downward trend in peak discharge with a distinct shift in the flood timing throughout the MRB. The outcomes of the study call for developing efficient adaptation strategies to ensure regional flood resilience since variations in the peak discharge timing should not be confounded with (insignificant) changes in its magnitude.

25 .



## 1 Introduction

In India, floods have caused an estimated average loss of US $54.3 billion between the years 1953 and 2016 (Chandra, 2019). Although the Intergovernmental Panel on Climate Change - Special Report on Extremes (IPCC SREX: Summary of policymakers, 2012) suggests, there has been a low agreement and

hence, less confidence in trends in flood magnitude or its frequency at the global scale, a recent study (Halgamuge and Nirmalathas, 2017) has indicated a moderate increase in floods and the number of affected people (~152k) in India over the past decades (1985-2016). Further, with increasing temperature, the severity and frequency of fluvial floods are expected to exacerbate in a changing climate (Hirabayashi et al., 2013; IPCC Working Group I: The Physical Science Basis, Stocker et al., 2013).

To date, several studies have analyzed temporal changes in the flood severity and persistence in its timing at global (Najibi and Devineni, 2018), continental (Cunderlik and Ouarda, 2009; Petrow and Merz, 2009; Burn et al., 2010, 2016; Hall et al., 2014; Blöschl et al., 2017; Matti et al., 2017; Burn and Whitfield, 2018; Mangini et al., 2018) and watershed scales (Jain and Lall, 2000; Tian et al., 2011; Bawden et al.,

2014; Jena et al., 2014 and Panda et al., 2013). The review of the literature shows, most of the previous studies are focused on either analyzing only monotonic (i.e., consistently increasing or decreasing; [Petrow and Merz, 2009; Hall et al., 2014; Blöschl et al., 2017; Matti et al., 2017; Mangini et al., 2018]) and abrupt (Nka et al., 2015; Villarini et al., 2009a) shifts in floods or seasonality of flood regimes (Cunderlik and Ouarda, 2009; Burn et al., 2010, 2016; Burn and Whitfield, 2018). Further, most of these

assessments cover either North America (Burn et al., 2016; Burn and Whitfield, 2018; Regonda et al., 2005) or Europe (Blöschl et al., 2017) focusing over floods across nival (i.e., short-duration but severe floods in April-May following spring thawing of winter snows) and mixed flow regimes in mid-(35-55° N) and high latitudes (> 60° N). Recently, Do et al. (2019) assessed seasonal timing of annual maxima floods over 7000 gauging stations globally using daily discharge record from 1981 - 2010. In India, Panda

et al. (2013) analyzed trends in seasonal and sub-seasonal streamflow over 19 stream gauges in the Mahanadi River Basin (MRB) for the period 1972 – 2007. Jena et al. (2014) analyzed 1957 – 2011 streamflow records at two of the gauging stations at the upper and middle reaches of MRB and have shown a significant increasing trend in floods. Jain et al. (2017) analyzed trends in peak floods over seven





major river basins in India using daily streamflow records from each of the basins. In their analysis, the
streamflow records vary between 15 and 48 years long during the 1965 – 2013 analysis period. One of
the primary findings of Jain et al. (2017) was decreasing or no trend in severe floods over most of the
gauges, which authors have attributed to multiple anthropogenic activities, such as land-use changes due
to increased urbanization, and control structures such as construction of diversion structures and
reservoirs. Their study also highlighted river regulations through reservoirs, have drastically reduced the
peak discharge magnitude over past 50 years that would have been increased under pristine conditions.
Ganguli et al. (2019) have shown sensitiveness of flood magnitude and persistence in the peak discharge
timing to the antecedent catchment wetness over MRB. However, to the best of our knowledge none of
the literature (Table S1; Panda et al., 2013; Jena et al., 2014; Jain et al., 2017; Ganguli et al., 2019; Do et
al., 2019) have considered coincidence of trends in peak discharge and its timing using observed
streamflow records from multiple gauging sites over a large river network in general and tropical pluvial
discharge regimes in particular.

To fill these gaps (Table S1), we analyze trends in flood magnitude and persistence in its timing on a
tropical pluvial river, MRB (Figure 1), in India. With a drainage area of 141 589 km$^2$, MRB contributes
a mean annual discharge of around 1895 m$^3$/s; out of which the maximum discharge during monsoon
season is around 6352 m$^3$/s and ranked second in the nation in flood-producing capacity (NIH, 2018;
Dadhwal et al., 2010). We have chosen MRB as a testbed since the review of the literature (Brakenridge,
2018) shows between 1985 and 2016 (within a span of 32 years), the MRB has experienced 25-major
flood episodes, causing immense losses to lives and properties; among which 22 events were due to
prolonged monsoon rainfall, extreme precipitation followed by dam releases and only 3 of them were
owing to tropical cyclones in the delta region. The historical records indicated MRB has experienced
severe floods in the past, *i.e.*, 2001, 2003, 2006, 2008, 2011, 2013, and 2016 (Beura, 2015; Jena et al.,
2014) and the latest flood event was between 26[th] April and 5[th] May 2019 (OSDMA, 2019). Also the
literature suggests, the frequency of occurrence of large floods is amplifying with an increase in extreme
precipitation and loss of forest cover in the lower and middle drainage basins of the River (Dadhwal et
al., 2010; Panda et al., 2013; Jena et al., 2014). In contrast, drainage basins in the upper reaches (above





the Hirakud dam) experience minimum flooding because of control structures, levees, and relatively steep slopes. A detection and attribution study (Mondal and Mujumdar, 2012) over basin-wide monsoon precipitation and streamflow at Hirakud reservoir have shown a marked contribution of human-induced

climate change to flow regimes rather than the observed precipitation pattern in the basin. Thus, understanding the basin-wide flood regime shift at densely populated MRB (Figure 1, b) holds a great interest.

Our analyses contribute to following aspects in understanding floods over MRB: *First*, we present an

assessment of spatial trends in peak discharge magnitude and persistence in its timings using two different flood samples, derived from the monsoonal maximum (MMF)- and peak over threshold (POTF) flood events, over 24 stream gauges over MRB during 1970 – 2016 (*i.e.*, 47-year long) time periods. *Second*, we assess the sensitivity of flood magnitudes to catchment size and mean basin elevation. Finally, we assess the coincidence of trends in peak discharge and shifts in flood timing during past four decades

(1970-2016). By leveraging in-situ observations, our analyses detect "flood rich and flood poor" regions (Merz et al., 2018) in MRB, which helps to underpin understanding of hydrologic responses to climate change under the effect of multiple stressors (Vano et al., 2015; Vano and Lettenmaier, 2014). The detected trends in flood discharge time series would help to analyze nonstationary flood frequency, useful for developing flood risk management portfolios in a changing climate (Hall and Solomatine, 2008;

Mirza, 2003; Rosner et al., 2014). Furthermore, the coincidence analysis of trends in floods and persistence in flood timing would able to identify potential hotspot locations, which can provide valuable information for decision support and cascade reservoir operations (Reddy and Kumar, 2007). Meanwhile, increasing vulnerability of water-related disasters (i.e., susceptibility to up/downward trends in fluvial discharge leading to high flows or floods or water-stress or drought conditions) and shifts in the timing

of streamflows in a changing climate could alter regional food-energy-water nexus. The methodology adopted here can be easily transferred to understand the nature of floods in similar climatic regions and analyze future climate projections. The outcomes of the research can provide a comprehensive understanding of the nature of floods in the basin in the present-day climate.



## 2. Data and Methods

**2.1 Study Area and Available Hydrometric Observations**

The Mahanadi River basin is mainly located in eastern part of India (from 80.50° - 86.83° E longitudes and 19.33° - 23.58°N latitudes) and flows through Chhattisgarh (52.42%), Odisha (47.14%), Maharashtra (0.23%), Madhya Pradesh (0.11%) and Jharkhand (0.1%) states. The river rises at Baster region in Dhamtari district of Chhattisgarh state. After flowing through an 851 km of length, it drains into the Bay

of Bengal. According to the 2011 Census (NIH, 2018; Census, 2011), 23.09 Million people (76% urban and 23% rural) resides in MRB. The rainfall in the basin mainly occurs during monsoon (June – September) season with the highest rainfall observe between  July and the first week of September. From December to January is the coldest period in the basin when the minimum temperature ranges from 4°C - 12°C. During summer (in May), the maximum temperature ranges between 42°C and 45°C (CWC,

2014). Based on basin morphology (Figure 1a), the MRB is divided into three distinct regions namely (CWC, 2014):

    a.   Upper (Region I): Drainage area between the source and the Hirakud dam also called the upper region. The area of this section is 84 700 km$^2$ out of which 75 136 km$^2$ lie in Chhattisgarh state.

    b.   Middle (Region II): Drainage area between Hirakud dam and head of delta also called the middle

125        region. The area of this region is 50 745 km$^2$.

    c.   Lower (Region III):  Drainage area between the head of delta and Bay of Bengal also called delta region.

Around 65% of the drainage basin is located in the upstream of the Hirakud dam (Dadhwal et al., 2010). The Hirakud dam is a multipurpose project intended for flood control, power generation, and irrigation.

It was built over MRB, approximately 15 kilometers upstream from Sambalpur in  Odisha state in 1957 with a catchment area of 83 400 km$^2$ (85% of catchment area lying in Chhattisgarh). The dam has a live storage capacity of 4823 million cubic meters and a spillway capacity of 42 450 m$^3$/s. Figure 1(b) shows the district wise population density highlighting major cities (Census, 2011) indicating the population varies between 1230 and 407 0000. The largest population is in Raipur (the state capital of Chhattisgarh)

followed by Durg and Bilaspur in Chhattisgarh state; Cuttack, Khordha, and Sundargarh in Odisha state.





## 2.2 River Discharge Data

We obtain daily streamflow records from 43 sites between 1971 and 2016 from India-Water Resources Information System (India-WRIS: http://indiawris.gov.in/wris/#/), Government of India. The sites contains varying record lengths; hence we select stations with more than 10 years of full records with at least 70% of discharge data available in monsoon months. After a pre-processing procedure, we finally selected 24 sites and exclude remaining stations containing large number of missing records. Table 1 and Figure 1a provide periods of data available and the locations of selected gauges over MRB. The MRB was delineated using a 90-m resolution Shuttle Radar Topography Mission - Digital Elevation Model (SRTM-DEM at 3 arc-seconds; Reuter et al., 2007) in Arc GIS version 10.1 software. The average basin elevations range between ~1300 m and 380 m above the mean sea level (MSL).

## 2.3 Method

We employ two methods to characterize floods, namely monsoonal (June-September) maxima flood (MMF) and Peak over Threshold flood (POTF) events. To detect monotonic and abrupt changes in the peak discharge time series, we apply Mann-Kendall (MK) trend and Pettitt change point tests. Trend analyses were conducted both at '*local* (at-site)' and '*regional*' (collection of sites) scales. The analysis of persistence in flood timing is carried out using circular statistics.

### 2.3.1 Extraction of Monsoonal Maximum (MMF) and Peak over Threshold Flood (POTF) Events

Typically floods in rainfed river basins in India are characterized by the maximum discharge during monsoon months in a calendar year (1 January –31 December). We select the maximum peak discharge during monsoon season for each year (*i.e.,* one event per year) from daily discharge record. In general, the POTF series provides more knowledge about statistical aspects of floods than that of the MMF events (Burn et al., 2016; Svensson et al., 2005), by selecting more than one peak discharge events per year, allowing better use of the available data. In contrast, choosing an appropriate threshold is a challenging aspect (Burn et al., 2016) for the extraction of the POTF time-series. Here, we evaluate different thresholds, varying from 98 to 99.9$^{th}$ percentiles at an increment of 0.5. Finally we select 98.5$^{th}$ percentile as a threshold level, enabling us to choose on an average three extreme events per year. To assure POTF





events are independent and identically distributed (*iid*), following earlier studies, we selected different time spans based on the catchment area of gauge stations (Petrow and Merz, 2009; Svensson et al., 2005; Ganguli et al., 2019): 5 days for catchment size below 45 000 km$^2$ and 10 days for catchment size ranges from 45 000 - 100 000 km$^2$. In our study, about 80% of stations have catchment areas less than 45 000 km$^{2,}$ and the remaining 20% of gauging stations have catchment areas varies between 45 000- and 100 000 km$^2$. If two successive POTF events occurred at a close time interval, the smaller event is discarded, and the higher one is picked up for further analysis.

**2.3.2 Detection of Trends and Change Points in Peak Discharge Events**

We apply the MK (Helsel and Hirsch, 2002) and the Pettitt (Pettitt, 1979) tests to detect monotonic trends and abrupt shifts in the peak discharge time series. The null hypothesis of the tests assumes that there is no trend or no change points exist in the peak discharge events at selected significance levels. The nonparametric rank-based MK test is often been used for evaluating monotonic trends in hydroclimatic time series (Burn et al., 2016; Petrow and Merz, 2009; Kunkel et al., 1999). However, the presence of autocorrelations in a time series can affect the the calculation of MK test statistics (Storch et al., 1999). Therefore, trends in MMF and POTF time series are assessed using the MK trend test accounting for auto-correlation and ties (Reddy and Ganguli, 2013) at a 10% significance level. The slope (change per day) of flood magnitude over the analysis period is determined using the Theil-Sen estimator for slope (Sen, 1968; Theil, 1950) estimate. The power of trend estimates at a '*local*' (i.e., stream gauge) level is evaluated using standard statistical significance tests, whereas trend at a '*regional*' scale is analyzed using field significance tests (at a 10% significance level). The field significance is assessed by False Discovery Rate (FDR) based test (Benjamini and Yekutieli, 2001), which is known to be relatively insensitive to spatial dependence among sites (Khaliq et al., 2009).

The change points in the peak discharge time series are assessed using abrupt changes in means (*i.e.*, average peak discharge) by applying the Pettitt test for single change-point detection (Villarini et al., 2012; Perreault et al., 2000; Villarini et al., 2009a). We calculate the power of the statistical test at three significance levels, $\alpha$ = 5%, 10%, and 15%. At $\alpha$ = 10% significance level, only 3 and 2 out of 24





stations had a change point for MMF and POTF events, respectively. Hence, we relaxed the significance

level up to $\alpha = 15\%$, to increase the statistical power of the test. All three statistical tests described here

are nonparametric, which are robust to the presence of outlier as well as the record length.

### 2.3.3 Persistence in Flood timing

The persistence in flood timing is analyzed using directional statistics (Mardia, 1972; Fisher, 1993;

Pewsey et al., 2013), which is a widely used method to define the timing (date of occurrence) and the

persistence of flood events (Burn et al., 2016; Cunderlik 2004; Tian et al., 2011). Using this method, first,

we express individual flood dates as a directional variable, and then compute directional mean and

variance from the record.

The flood date (in Julian date $(JD)_i$) is converted to an angular value $(\lambda_i)$, (in radians) using following

expression (Laaha and Blöschl, 2006)

$$\lambda_i = JD_i \frac{2\pi}{l(yr)} \tag{1}$$

Where, $i$ shows peak discharge events, $JD$ indicates Julian dates ranging from 1 to 365 or 366 days for

31[st] December considering a non-leap or a leap year; $l(yr)$ denotes the number of days in a calendar year,

i.e., 365 days accounting for a normal year, whereas 366 days for a leap year. For $n$ flood events, we

determine the mean flood date from the following equations (Burn and Whitfield 2018)

$$\overline{x} = \frac{\sum_{i=1}^{n} q_i \cos \lambda_i}{\sum_{i=1}^{n} q_i} \; ; \overline{y} = \frac{\sum_{i=1}^{n} q_i \sin \lambda_i}{\sum_{i=1}^{n} q_i} \tag{2}$$

where, $\overline{x}$ and $\overline{y}$ denote the x- and y-coordinates of the mean event date. Following Burn and Whitfield

(2018), we obtain the equation from weighted average of the sampled extremes by weighing the

corresponding peak discharge. The occurrence time of a flood in a year is derived from the mean event

angle, $\overline{\lambda}$ using following expressions



$$\bar{\lambda} = \begin{cases} \tan^{-1}\left(\dfrac{\bar{y}}{\bar{x}}\right), & \text{if } \bar{x} > 0 \text{ and } \bar{y} > 0 \\[2ex] 180 - \tan^{-1}\left(\dfrac{\bar{y}}{\bar{x}}\right), & \text{if } \bar{x} < 0 \text{ and } \bar{y} > 0 \\[2ex] 180 + \tan^{-1}\left(\dfrac{\bar{y}}{\bar{x}}\right), & \text{if } \bar{x} < 0 \text{ and } \bar{y} < 0 \\[2ex] 360 - \tan^{-1}\left(\dfrac{\bar{y}}{\bar{x}}\right), & \text{if } \bar{x} > 0 \text{ and } \bar{y} < 0 \end{cases} \tag{3}$$

The mean flood date is determined using following expression (Laaha and Blöschl, 2006)

$$\delta = \bar{\lambda} \times \left(\frac{l(yr)}{2\pi}\right) \tag{4}$$

Where $\delta$ is the mean date of flood occurrence. The persistence in the timing of flood, $\bar{r}$ is determined from the mean resultant length, which is given by (Laaha and Blöschl, 2006)

$$\bar{r} = \sqrt{\bar{x}^2 + \bar{y}^2}, \qquad 0 \le \bar{r} \le 1 \tag{5}$$

where, '$r$' indicates persistence in the flood timing, which is a dimensionless number. When $\bar{r} = 0$, $\bar{\theta}$ is not defined. This situation arises when occurrence of peak discharge events are uniformly distributed throughout over a year, indicating no persistence in flood timing. On the other hand, $\bar{r} = 1$ shows all floods at a site occur on the same day of the year indicating high persistence in the flood timing. The average date of flood may fall at a period of the year even when no peak discharge occurred in a particular year (Burn and Whitfield, 2018). The long-term temporal evolution of the variability in peak discharge events is computed using the circular variance, $\sigma^2$ (Dhakal et al., 2015)

$$\sigma^2 = -2\ln(\bar{r}) \tag{6}$$





The trend in the timing of floods is estimated using a nonparametric adjusted Theil-Sen estimator for slope. The slope estimator $\beta_{circular}$ shows the median of the difference in flood dates over all possible combination of years ($i$ and $j$) within the time series, which is given by

$$\beta_{circular} = med\left(\frac{JD_j - JD_i + k}{j - i}\right) \qquad (7)$$

with

$$k = \begin{cases} -\bar{m} & if & (JD_j - JD_i) > \bar{m}/2 \\ \bar{m} & if & (JD_j - JD_i) < -\bar{m}/2 \\ 0, & & otherwise \end{cases} \qquad (8)$$

where

$$\bar{m} = \frac{1}{n}\sum_{i=1}^{n}(l(yr))_i \qquad (9)$$

where $med(\bullet)$ denotes median and $k$ adjusts to the circular nature of the dates and $\beta_{circular}$ has units of days per year.

## 3. Results and Discussion

### 3.1 Trends in Peak Discharge over Past Four Decades

Analyzing changes in floods is vital in many hydrological applications, such as hydropower production, and further could lead to improved flood frequency estimates in a nonstationary climate (Kwon et al., 2008; López and Francés, 2013; Villarini et al., 2009b; Vogel et al., 2011). We detect monotonic trends in peak discharge series using the MK trend test accounting for autocorrelation and ties in the time series. Figure 2 shows the spatial trends in MMF and POTF flood time-series for all 24-gauges in MRB. Table 2 presents values of MK test statistics, associated p-values of the test and the estimated slopes at individual stream gauges. While none of the monsoonal maxima flood series (Figure 2, a) showed a local significant increasing trend (at 10% significance level), the POTF series revealed significant increasing trends at Kurubhata, Hirakud Dam and Kantamal gauging stations (Figure 2, b). In both MMF and POTF flood samples, Region I exhibited a mix of up- and downward trends with the latter dominating the former,





although many of these trends are statistically insignificant. This agrees with Jain et al. (2017), in which

authors noted a decreasing or no trend of severe floods in some of the large river basins in India, including

MRB. However, their analysis was based on only one gauging station from each of the basins, out of

which Tikarapara is in MRB. Second, their analysis period was limited to only 40 years (1971 – 2010)

streamflow records. The decreasing trends in the peak discharge time series might be a consequence of

changing rainfall patterns and significant decreasing rainfall trends over the basin, especially during the

month of August (Panda et al., 2013). Further, the streamflow at most of the sites upstream to the Hirakud

reservoir can be considered unregulated owing to a lack of major control structures upstream of the

Hirakud reservoir (Panda et al., 2013).

Most of the stations in Region II, except Tikarapara showed increasing trends (either significant or

insignificant). For example, despite the presence of dense deciduous vegetative cover, an increasing trend

in flood magnitude at Kantamal sub-basin could be attributed to deforestation at the upstream sub-basins

as reported in an earlier study (Mishra et al., 2008). Also, Jena et al. (2014) have shown that the recent

(post-1957, the year of Hirakud dam construction) incidence of large floods is the consequence of an

increase in extreme precipitation in the middle part of the drainage basin. An insignificant decreasing

trend at Tikarapara gauge station is due to the dense vegetative cover, and the medium-textured soil type

over the watershed area, which amplifies the infiltration processes and hence reducing runoff over the

basin (CWC, 2014). An insignificant increasing trend at Naraj station could be attributed to its proximity

(the distance between Naraj gauging station and the coast is only 120 km) to the Bay of Bengal coast and

high-tide induced flooding across the delta that could extend inwards due to small width of the delta

(Choudhury et al., 2012). As extreme sea level resulting from severe storm surges across the Bay of

Bengal (Flierl and Robinson, 1972; Milliman et al., 1989) pushes ocean tides upstream, the tidal signal

propagates from river estuaries to inland, which in turn can amplify the flood hazard in delta region

(Ensign and Noe, 2018; Lyddon et al., 2018; Ganguli and Merz, 2019). The floods in the low-lying deltas

can stem from the superposition of storm surges and river floods, which is often associated with common

meteorological drivers, such as synoptic low-pressure systems and extreme precipitation. The moisture-

laden air-mass from coast moves inland over the catchment leading to heavy and persistent rainfall,





causing runoff-induced extreme flooding. Our results agree with Panda et al. (2013), in which authors have detected a significant upward trend in rainfall across the southwestern and coastal parts of the basin. The synoptic disturbances over the Bay of Bengal, leading to flash flood generating extreme rainfalls

(Panda et al., 2013) could be the potential causes of increasing flood trends in this region. We then apply a field significance test to understand regional behavior of flood trends, which indicated a significant downward trend over Region I for MMF events at a 10% significance level. On the other hand, the middle reaches of the basin does not show any field significant up- or downward trend. Furthermore, overall, we could not detect the presence of any field significant trend for the POTF series over the regions. Table S2

presents the results of the field significance test for both regions.

Next, we detect abrupt changes in the flood time series using the Pettitt test. Change points in a long historical data arises due to multiple consequences such as a shift in gauge location, land use and/land cover changes, reservoir regulations, as well as climatic variations (Villarini et al., 2009a). Our results

show only a few stations exhibit change points (in location or mean of the distribution) in both MMF and POTF series (Figure S1). In MMF events (Figure S1, a), only 3 (16.67% of all the stations over MRB) out of 24 stations in the basin show statistically significant change points at a 10% significance level. Andhiyarkore shows the presence of a significant change point at a 15% significance level. For POTF events (Figure S1, b), Bamnidhi and Manendragarh exhibit abrupt changes at a 10% significance level

while Kurubhata shows a significant change point at a 15% significance level. None of the stations showed a change point at a 5% significance level. A closer look reveals that all three stations are spatially distributed over the northern part of the basin at Region **I**. It was observed that all these stations (*i.e.*, Andhiyarkore, Bamnidhi, Ghatora, Kurubhata, and Manendragarh) had a change point during the end of the 20[th] and beginning of the 21[st] centuries. However, our analysis could not detect any discernible spatial

patterns based on the detected change points with respect to the mean. Further, the detected change points at Bamnidhi (1987 and 1978 for MMF and POTF events, respectively) and Manendragarh (2003 and 2004 for MMF and POTF events, respectively) are close to the year of construction of 'mega-dams' (1985, 1976 and 2004) with a height > 15 m (Best, 2019) up-streams to the respective gauge stations (WRIS, 2015). Details of these dams, their capacity, and the corresponding gauge locations are presented



in Table S3.  For remaining stream gauges, we could not identify any specific reasons for the abrupt changes.

To summarize, our findings suggest the following: (i) little changes in flood magnitude during the 1970–2016-time window throughout MRB except for a few instances of locally significant up/downward trends

and change points at individual stream gauges. Further, we note evidence of decreasing field significant trend at Region I. (ii) We find a contrasting response of spatial trends in flood magnitude to the nature of flood series. For MMF events, ~ 67% (12 out of 18 stations) gauge stations showed (a mix of significant and insignificant) downward trends in flood magnitude at Region I, whereas 67% (4 out of 6 stations) of sites showed (insignificant) increasing trend at Region II. On the other hand, for POTF events, at Region

I, ~ 56% (10 out of 18 stations) of sites showed spatially coherent (a mix of significant and insignificant) decreasing trends, whereas, at Region II, 80% (4 out of 5 stations) gauge stations showed (a mix of significant and insignificant) increasing trends in flood magnitude. Overall, a larger number of sites show upward trends in flood magnitude for POTF events (52% stations) than that of the MMF event (42% stations). (iii)  The change point analysis of flood series suggests only ~ 13 (POTF) – 17 (MMF) % of the

stations show change points over the entire river basin.

## 3.2 Scale-Dependency in the Nature of Flood Trends

Next, we analyze the influence of scale-dependency on trends in peak discharge. The objective of the scale-dependency test is to investigate if the nature of changes in streamflow patterns is related to the

catchment size and elevation of the basins. For this, trends in peak discharge magnitude of each flood samples are plotted against the mean basin elevation (Table 1) and catchment area. Figure 3 presents three-dimensional plots to analyze the role of catchment size and mean basin elevation on trends in flood magnitude. The significant changes are marked (Figure 3) in solid. We observe no clear pattern of scale-dependency (either scales or elevations), where significant changes are concentrated. However, for MMF

events, decreasing trends are more concentrated between catchment area 1000 and 3000 km$^2$, and the mean elevation between 490 and 700 m MSL (Figure S2). However, none of these downward trends are statistically significant. On the other hand, a few evidences of significant increasing trends (at three of the





gauging stations, Kurubhata, Kantamal, and Hirakud dam) are observed for larger catchments (catchment area more than 1000 km$^2$) with higher basin elevation (ranges between 250 and 500 m MSL) for the POTF
flood events (Figure S2). Despite being located at a higher elevation, the downward trend in Manendragarh station for both flood series is attributed to the construction of mega-dams upstream in 2004 (Table S3).

**3.3 Persistence in Flood Timing**

We determine the persistence in peak discharge timing using directional statistics. The mean flood date, temporal variability $\left(\sigma^2\right)$ and the persistence $\left(\bar{r}\right)$ in flood timing are evaluated for both methods of flood samplings at respective stream gauges (Figure 4). For the majority of sites, the mean flood dates are temporally concentrated around the mid of August for both MMF and POTF series (Figure 4, a – b) except for Andhiarkore and Naraj, which showed the mean flood dates are close to September, *i.e.*, 29$^{th}$ August
and 27$^{th}$ August for the MMF events (Figure 4, c). While Andhiarkore is located at Region I, in upper MRB, Naraj is located in the delta region. Further, we observe that considering both methods of flood samples, the POTF series shows a stronger tendency in temporal clustering of mean flood dates around mid of August (Figure 4, b-c) and higher circular variance than that of the MMF series (Figure 4, d). Overall, we infer that the timing of floods is highly persistent across MRB. Our results agree with an
earlier assessment (Burn and Whitfield, 2018), which have shown that the pluvial discharge regime, typically, show a larger persistence in peak discharge timing as compared to other flow regimes. Further, studies (Blöschl et al., 2019; Villarini, 2016; Do et al., 2019) have suggested, unlike trends in flood magnitude or frequency, trends in average flood timing remains relatively independent of anthropogenic influences, for example, land use, constructions of dams or reservoirs, while geographical locations of the
drainage basins play a prominent role in modulating the persistence in flood timing.

**3.4 Shifts in the Timing of Peak Discharge**

The shifts in the timing of peak discharge events are analyzed using an adjusted Theil - Sen slope estimator with the correction for the circular nature of mean flood dates. Figure 5 shows trends in the timings of





flood events in MRB. While the location of the stream gauges is represented using a square box, the hues

in red and green of the boxes indicate the gauges with earlier and later occurrences of peak discharge

events. The stream gauge with color in white represents the flood events that have concurred with the

mean flood date at respective stream gauge locations, whereas darker shades show a substantial deviation

from the mean flood date.


For the MMF events (Figure 5, a), most of the stations in Region II showed an earlier occurrence of peak

discharge events. At the delta region, Naraj site, which is located at a relatively low elevation (Table 1),

shows a delay in flood timing up to 18 days (per decade). While Manendragarh gauging site is located at

a relatively higher elevation (Table 1), experience an earlier occurrence of floods. At Region II, in contrast

to MMF events, POTF events show no shifts in flood timings for most of the gauges (Figure 5, b).

However, we note a delayed occurrence of peak discharge at Kesinga. A few gauging stations show

disparate trends in flood timings, with an earlier occurrence of floods at MMF events while delayed

occurrence for the POTF events and vice versa. For both methods of flood samplings, the Hirakud dam

showed a delayed occurrence of flood, whereas, Tikarpara near the delta showed an earlier occurrence of

floods. The delay in flood occurrence in Hirakud could be attributed to a shift towards greater water

inflow during September in the Hirakud reservoir in recent decades. This is in agreement with Choudhury

et al. (2012), in which authors report a delay in peak discharge events in the Hirakud reservoir based on

the analysis of mean inflow trends using available discharge records covering the period 1958-2008. The

delayed flood timing in Hirakud has critical implications for flood control operation of the dam since, by

the monsoon end (*i.e.*, during September), dam management authorities are in a dilemma whether to open

the reservoir gate. For instance, severe floods in the downstream of Hirakud during the year 2001, 2008

and 2011 (Brakenridge, 2018; Jena et al.*,* 2014) were triggered by the delayed release of water from

several full dams coupled with heavy rainfall during monsoon months. On the other hand, an earlier flood

occurrence at Tikarpara could be attributed to the loss of forest cover area across the middle reaches of

the basin (Dadhwal et al., 2010). Further, Dadhwal et al. (2010) highlighted that the decrease in forest

cover by about 5.7% at Region II has resulted in an increase in fluvial discharge by approximately 4.5%

at Munduli in the delta region, which is located at 4.5 km upstream to Naraj. Furthermore, the heavy





rainfall induces a massive inflow in the Hirakud reservoir leading to the dam authorities to open gates of the dam, causing flash floods in the delta region (Panda et al., 2013).


At Region I, ~56% (10 out of 18) and 61% (11 out of 18) of the stations show delay in flood occurrences for both MMF and POTF events respectively, while at Region II, ~ 67% (4 out of 6) exhibits earlier dates of floods for MMF events, whereas, no shifts in flood timing is observed for POTF events at 60% of (3 out of 5) stations. Overall, our results suggest a greater number of stream gauges show delayed timing in 390 flood occurrences for both methods of flood samplings.

## 3.5 Coincidence of Trends in Peak Discharge and Persistence in its Timing

It is interesting to note that in both methods of flood samplings, we find sites with upward (downward) trends in flood magnitude coincides with delayed (earlier) occurrences of floods. Figure 6 suggests flood 395 risk assessments would need to consider relatively more vulnerable sites that show either down or upward trends in flood magnitude coincide with an earlier occurrence of floods. Although the nature of trends in flood magnitude may be insignificant for a specific site, when it concurs with an early or delayed occurrence, the resulting impact would be catastrophic to the exposed population, since they may not be prepared for such events (AghaKouchak et al., 2018). Our analysis shows downward trends in flood 400 magnitude at both Bamnidhi and Manendragarh are significant at a 5% significance level (Table 2); however, floods are expected to occur early (Figure 6) in these sites. A closer look reveals, most of the sites located in quadrant I (i.e., a decreasing trend in flood magnitude with an earlier occurrence of the flood) are spatially clustered across Region I except Tikarpara. Considering MRB as a whole, ~33% (in MMF) and ~17% (in POTF) of sites unveiled a downward trend in flood magnitude with an earlier date 405 of flood occurrence. On the other hand, ~29% - 35% of the stations show an upward trend in flood magnitude with a delayed shift in flood timings. Table 3 summarizes the list of such stations.

Although detected trends in flood magnitude (Figures 2, S1-S2) and shifts (delayed or earlier) in timing of floods (Figure 5; Table 3) over MRB could be linked to human intervention (Best, 2019; Choudhury 410 et al., 2012; Dadhwal et al., 2010; Mishra et al., 2008), rainfall-induced runoff during monsoon season is

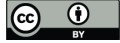



the prominent flood generating driver over MRB (Jena et al., 2014; Panda et al., 2013). Due to extreme precipitation during monsoon season, the catchment rapidly saturates, resulting in floods over MRB. One of the possible physical mechanisms is to link extreme precipitation events to an increase in the magnitude of fluvial floods in a warming world. Despite most of the sites in the middle reaches show (a mix of significant and insignificant) upward trend in flood magnitude, our results suggest a relatively regular peak discharge (persistence index ranges between 0.88 and 0.95 for MMF events and 0.87 and 0.93 for POTF events) with mean date of flood occurrences during the mid of August for most of the sites. Our findings suggest that floods are mostly affected by changes in flood generating processes, such as available antecedent moisture content in the catchment (Ganguli et al., 2019) and shift in atmospheric circulation pattern that results into changes in dominant storm mechanisms, rather than the increase in precipitation extremes in response to increase in surface warming (Wasko et al., 2019; Sharma et al., 2018; Wasko and Sharma, 2017). The review of the literature suggests (Pattanayak et al., 2017) a linkage between large-scale atmospheric teleconnections (such as the sea surface temperature, SST) and basin hydroclimatology (i.e., precipitation, temperature minima, and temperature maxima) over MRB. A strong linkage was identified especially during the 1980s, which was primarily attributed to changes in Pacific Decadal SST teleconnection patterns and anthropogenic influences (Pattanayak et al., 2017).

Although we find a spatially coherent flood pattern in MRB using both methods of flood samplings, the nature of trends differs on a case to case basis. This is because, while in monsoonal maxima event, we select the maximum peak flow of each year during monsoon months (i.e., June to September), POTF samples are extracted from all peak values from the total time series that lie above a certain truncation level ensuring *iid* nature of the selected flood events. Thus, a very low discharge value, especially during a dry period, can also become an MMF event, whereas, some peaks that are not maximum but still with a very high discharge value could be included as a POTF event (Bezak et al., 2014).

## 4. Concluding Remarks

Analyzing changes in flood magnitude and shifts in flood timing can provide valuable insights regarding water availability (high and low flows), managing water security, and help in assessing fidelity of





hydrological models, thereby encouraging better management policies for regional water resources sustainability. This paper contributes to the comprehensive assessment of trends in flood magnitude and
shifts in the timing of floods in densely populated Mahanadi river basin. Although there are several studies in the literature (Table S1) that analyses trends in either severity or timing of floods alone at a regional or global scale, to the best of authors' knowledge, no studies so far have analyzed both magnitude and timing of floods and their coincidence especially in a tropical pluvial large river basin system. Further, the most detrimental human impacts owing to large floods could be potentially from developing countries because
of low flood protection strategies and vulnerability of populations (Tessler et al., 2015). Using 47-year (1970 – 2016) streamflow records and two different sampling methods of peak discharge events, this study for the first time assesses changes (both monotonic and abrupt) in flood magnitude and shifts in the timing of floods in densely populated MRB, which is home to the longest dam in India, Hirakud dam and reservoir systems, and lifeline to around 71 million people residing in this region (Pattanayak et al., 2017).

The novel insights from the study are worth highlighting:

- A spatially coherent pattern of the flood is observed over MRB using both methods of flood samplings - Region I shows a mixture of (insignificant) increasing and (a mix of insignificant and significant) decreasing trend, in which the number of gauges with downward trends is more pronounced than that
of the number gauges with upward trends. Further, the downward trends in flood magnitude at Region I are field significant. Also, while we find a few evidence of (significant) change points at Region I, no such change points are detected across Region II.

- Except for the construction of a few major dams upstream that have affected the nature of flood trends, overall, we observe no clear linkage between flood severity and catchment morphology (i.e., mean
basin elevation and watershed area). For MMF events (insignificant) most of the downward trends (out of which two of them are statistically significant) in flood magnitude are clustered for gauges with catchment area between 1000 and 3000 $km^2$ and the mean basin elevation ranges from 490 to 700 m MSL. On the other hand, for the POTF events, we find a few evidence of a significant upward trend for larger catchment (of an area more than 1000 $km^2$) with high basin elevation (between 250
and 500 m MSL). Despite being located at a high elevation (668 m above mean sea level) with a small



catchment area (~ 1000 km$^2$), Manendragarh, which is located at Region I, showed a downward trend in flood magnitude for both MMF and POTF events. This could be a consequence of the construction of a dam near this site.

- The persistence analysis of flood timing using both methods of flood samplings suggests the mean date of floods are concentrated around the mid of August for most of the stream gauges. Although we note the nature of the mean dates of flood occurrences is more or less uniform across both methods of flood samplings, the POTF events show relatively high circular variance than that of the MMF events.

- Around half of the stream gauges across MRB exhibit delay in the date of flood occurrences. In addition, about ~29% (~33%) and ~33% (~17%) of the stations indicate an upward (downward) trend but with a delay (early) in the occurrence of floods for MMF and POTF events, respectively.

A few caveats could be considered. Our analysis is based on station-based observations. The derived insights are based on the quality of in-situ observations, which is limited to the recent four decades. Although many studies (Rao, 1993; Panda et al., 2013; Jena et al., 2014; Jain et al., 2017) so far analyzed the influence of hydro-meteorological drivers, precipitation and air temperature in generating peak discharge over MRB, no studies, to date have analyzed the role of catchment wetness in simulating the nature of floods. However, extreme rainfall does not necessarily translate into fluvial floods (Sharma et al., 2018; Wasko and Sharma, 2017; Wasko et al., 2019), antecedent wetness conditions often strongly controls the nature of flood peaks across the river basin (Merz et al., 2018). Hence, future research will be directed towards understanding the role of catchment processes, such as antecedent moisture content, in regulating the nature of floods in a large tropical pluvial flow regime, such as in MRB.

*Data Availability.* The daily streamflow discharge data are obtained from the Water Resources Information System (India-WRIS: http://indiawris.gov.in/wris/#/) maintained by the Central Water Commission (CWC), Government of India.

*Author Contributions.* PG developed codes, designed the problem and conceptualized the methodology. YRN collated the data with help from CC. YRN performed the analysis of trends and seasonality, except



the analyses presented in section 3.5, and prepared an initial draft. All authors discussed and interpreted the results. PG performed the analyses in the section 3.5 with input from YRN, and prepared the final draft of the manuscript with contributions from all co-authors.

*Competing Interests.* The authors declare no conflict of interest.

## Acknowledgments

The work was primarily funded by two Department of Science and Technology (DST) research grants, namely, the IFR (grant number DST/CCP/CoE/79/2017), "Impact of Climate Change on Flood Risk" as part of the Centre of Excellence in Climate Change Studies established at IIT Kharagpur and the HLR (grant number SRG/2019/000044), "Hydrological Drought Co-incidence Risk Analysis over Large River Basins in India". Ministry of Human Resources Development (MHRD), Government of India provided partial funding. Station-based daily streamflow data of Mahanadi River Basin are obtained from India-WRIS website.

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





**Table 1**. List of discharge gauging stations and period of data availability in MRB

| Gauging Station | Latitude (º) | Longitude (º) | Length of Records | Mean Basin Elevation (above MSL in m) | Catchment Area (km$^2$) |
|---|---|---|---|---|---|
| Andhiarkore | 21.83 | 81.61 | 1978-2013 (36) | 265 | 2210 |
| Bamnidhi | 21.82 | 82.63 | 1971-2013 (43) | 223 | 9730 |
| Baronda | 20.67 | 81.82 | 1977-2013 (37) | 290 | 3225 |
| Basantpur | 21.56 | 82.85 | 1971-2013 (43) | 206 | 57 780 |
| Ghatora | 22.06 | 82.22 | 1980-2013 (32) | 246 | 3035 |
| Hirakud Dam | 21.52 | 83.85 | 1991-2011 (21) | 175 | 83 400 |
| Jondhra | 21.60 | 82.36 | 1979-2013 (34) | 219 | 29 645 |
| Kantamal | 20.65 | 83.73 | 1984-2011 (40) | 118 | 19 600 |
| Kelo at Raigarh | 21.89 | 83.40 | 1996-2014 (18) | 230 | 950 |
| Kesinga | 20.20 | 83.23 | 1979-2011 (33) | 166 | 11 960 |
| Kotni | 20.91 | 80.89 | 1978-2013 (34) | 283 | 6990 |
| Kurubhata | 21.99 | 83.20 | 1978-2011 (34) | 215 | 4625 |
| **Manendragarh** | 22.93 | 82.25 | 1989-2013 (25) | **411**[*] | 1100 |
| **Naraj** | 20.48 | 85.76 | 2000-2011 (12) | **30** | 133 230 |
| Paramanpur | 21.79 | 84.11 | 2001-2013 (12) | 210 | 2120 |
| Pathardhi | 21.34 | 81.56 | 1989-2013 (23) | 280 | 2511 |
| Rajim | 20.97 | 81.88 | 1971-2013 (43) | 284 | 8760 |
| Rampur | 21.43 | 82.48 | 1971-2013 (43) | 232 | 2920 |
| Salebhata | 20.98 | 83.54 | 1973-2010 (38) | 140 | 4650 |
| Seorinarayan | 21.60 | 82.56 | 1986-2013 (28) | 209.5 | 48 050 |
| Simga | 21.63 | 81.69 | 1971-2013 (43) | 254.46 | 30 761 |
| Sukma | 20.78 | 83.37 | 1989-2002 (14) | 157 | 1365 |
| Sundergarh | 22.12 | 84.01 | 1978-2011 (34) | 214 | 5870 |
| Tikarapara | 20.56 | 84.52 | 1972-2016 (44) | 50 | 124 450 |

[*]Stream gauges with the highest and the lowest mean basin elevations are marked in bold letters.






**Table 2.** Analysis of monotonic trends in flood magnitude

| Gauging station | MMF Events | | | POTF Events | | |
|---|---|---|---|---|---|---|
| | MK test statistics | p – value[*] | Sen slope (day$^{-1}$) | MK test statistics | p – value[*] | Sen slope (day$^{-1}$) |
| Andhiarkore | -0.97 | 0.33 | -2.09 | 0.78 | 0.43 | 0.30 |
| **Bamnidhi** | **-3.73** | **0.00** | **-79.58** | -1.56 | 0.12 | -10.83 |
| Baronda | 0.60 | 0.55 | 13.95 | 1.26 | 0.21 | 1.98 |
| Basantpur | -1.26 | 0.21 | -94.00 | -0.93 | 0.35 | -14.06 |
| Ghatora | -1.11 | 0.27 | -4.96 | 0.61 | 0.54 | 0.66 |
| **Hirakud Dam** | 0.52 | 0.60 | 50.75 | **2.00** | **0.05** | **179.45** |
| Jondhra | -0.97 | 0.33 | -32.44 | -1.11 | 0.27 | -8.30 |
| Kantamal | 1.00 | 0.32 | 66.99 | **1.66** | **0.10** | **21.94** |
| Kelo | -0.41 | 0.68 | -2.64 | -0.06 | 0.96 | -0.68 |
| Kesinga | 0.73 | 0.47 | 63.81 | 0.21 | 0.83 | 1.91 |
| Kotni | 0.45 | 0.65 | 9.33 | 0.48 | 0.63 | 2.42 |
| **Kurubhata** | 1.07 | 0.29 | 8.67 | **1.82** | **0.07** | **3.21** |
| **Manendragarh** | **-2.41** | **0.02** | **-12.10** | **-1.99** | **0.05** | **-0.84** |
| Naraj | 1.17 | 0.24 | 1244.17 | - | - | - |
| Paramanpur | -0.31 | 0.76 | -65.41 | -0.21 | 0.84 | -24.43 |
| Pathardhi | -0.03 | 0.98 | -0.97 | -1.29 | 0.20 | -4.97 |
| Rajim | 0.21 | 0.83 | 8.47 | -0.49 | 0.62 | -2.43 |
| Rampur | -0.25 | 0.80 | -6.05 | 0.39 | 0.69 | 1.06 |
| Salebhata | 0.36 | 0.72 | 11.94 | 0.80 | 0.42 | 1.53 |
| Seorinarayan | 1.17 | 0.24 | 140.56 | 0.04 | 0.97 | 2.24 |
| Simga | -0.10 | 0.92 | -6.19 | -0.31 | 0.76 | -3.02 |
| Sukma | -0.77 | 0.44 | -27.36 | 1.19 | 0.23 | 4.81 |
| Sundergarh | -0.77 | 0.44 | -8.44 | -0.51 | 0.61 | -0.61 |
| Tikarapara | -1.62 | 0.11 | -168.87 | -0.68 | 0.50 | -57.56 |

[*]p $\leq$ 0.10 are marked in bold letters; p-values are rounded up to two significant digits. The number of POTF events is found to be less than 10 at Naraj, hence POTF events for this site are excluded from the analysis.



**Table 3**. Potential hotspots with a downward (upward) trend in flood magnitude coincided with an earlier (delayed) date of flood occurrence

|  | **MMF Events** | **POTF Events** |
|---|---|---|
| A downward trend in flood magnitude coincide with an early date of flood occurrence | Bamnidhi | Kelo at Raigarh |
|  | Ghatora | Pathardhi |
|  | Kelo at Raigarh | Simga |
|  | Manendragarh | Tikarapara |
|  | Paramanpur |  |
|  | Rampur |  |
|  | Sukma |  |
|  | Tikarapara |  |
| An upward trend in flood magnitude coincide with a delayed date of flood occurrence | Baronda | Andhiyarkore |
|  | Hirakud dam | Baronda |
|  | Kantamal | Ghatora |
|  | Kurubhata | Hirakud dam |
|  | Naraj | Kesinga |
|  | Rajim | Kotni |
|  | Seorinarayan | Kurubhata |
|  |  | Seorinarayan |






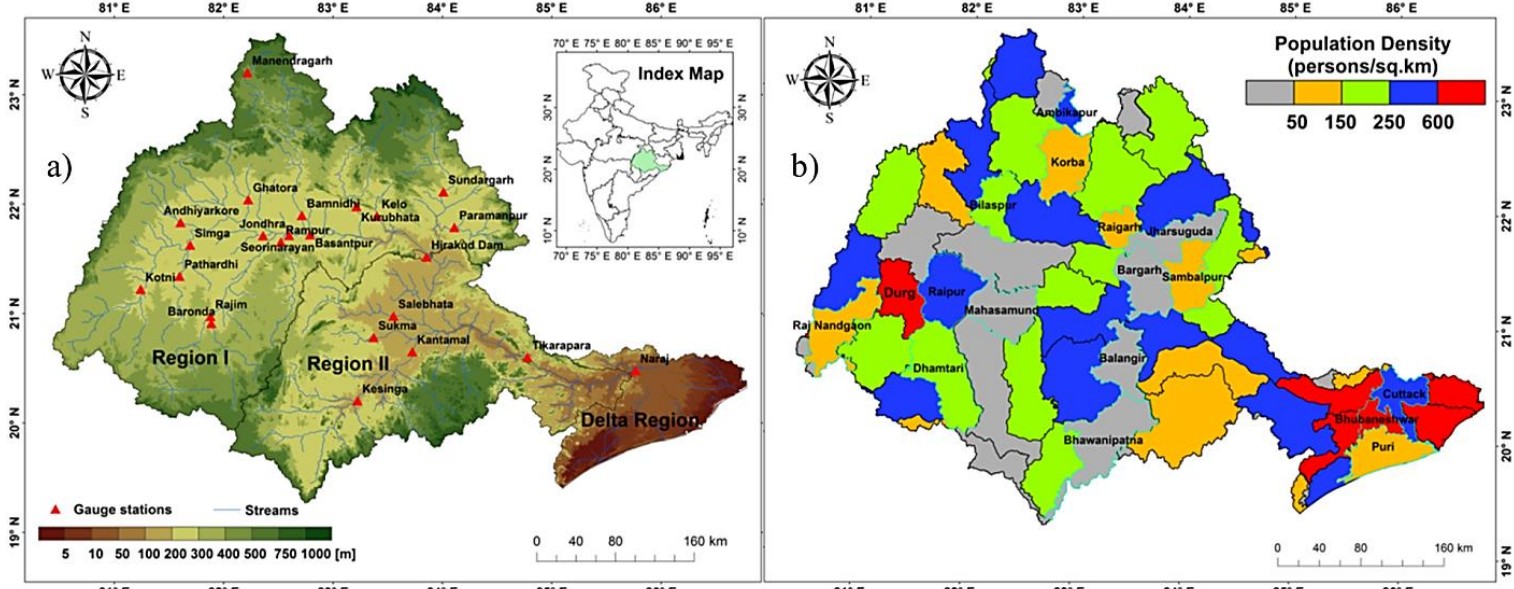

**Figure 1**. Mahanadi river basin (a) Elevation map (b) District wise population density with highlighting major cities
(Census, 2011).






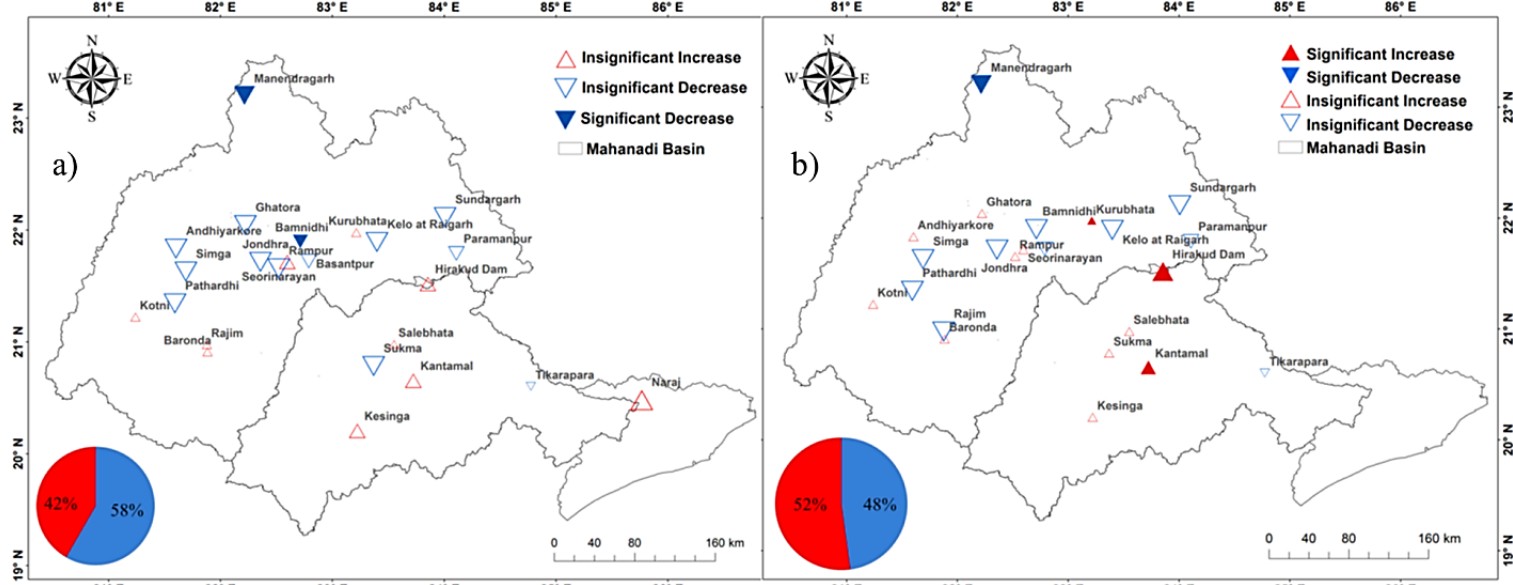

**Figure 2**. Trends in Flood magnitude over MRB for (a) MMF and (b) POTF events. The significance of the trend is evaluated at a 10% significance level. The size of a triangle is proportional to the slope (Table 2) obtained from Theil Sen slope estimator. The pie chart at the bottom left corner shows the percentage of gauges with increasing and decreasing trends in flood magnitude irrespective of their level of significance.


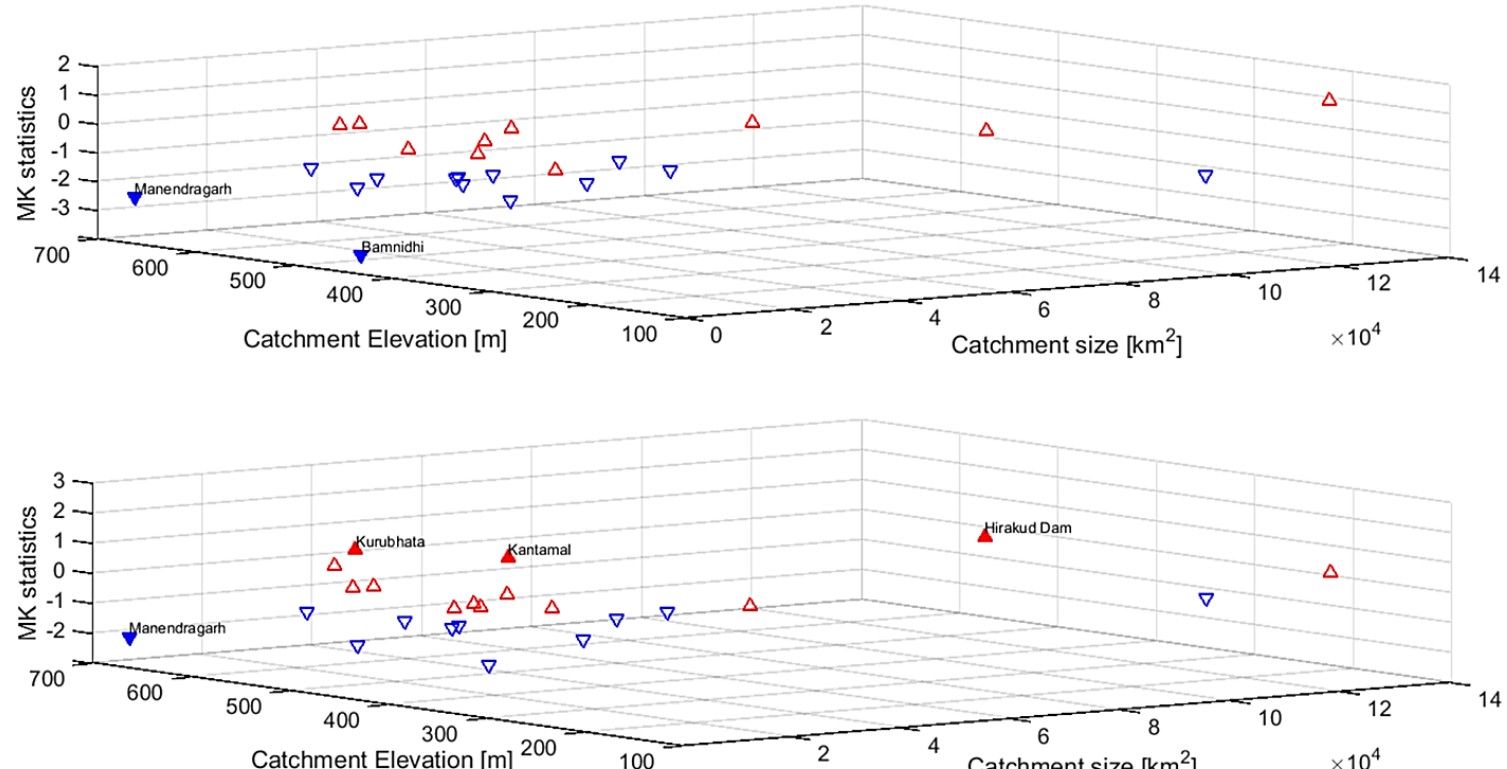

**Figure 3**. Scale dependencies relative to flood trends of MMF events (*top panel*) and POTF events (*bottom panel*). Mann-Kendall trend statistics are plotted as a function of the Catchment size (km$^2$) and Catchment elevation (m above MSL). Shaded triangles indicate significant changes at a 10% significance level.




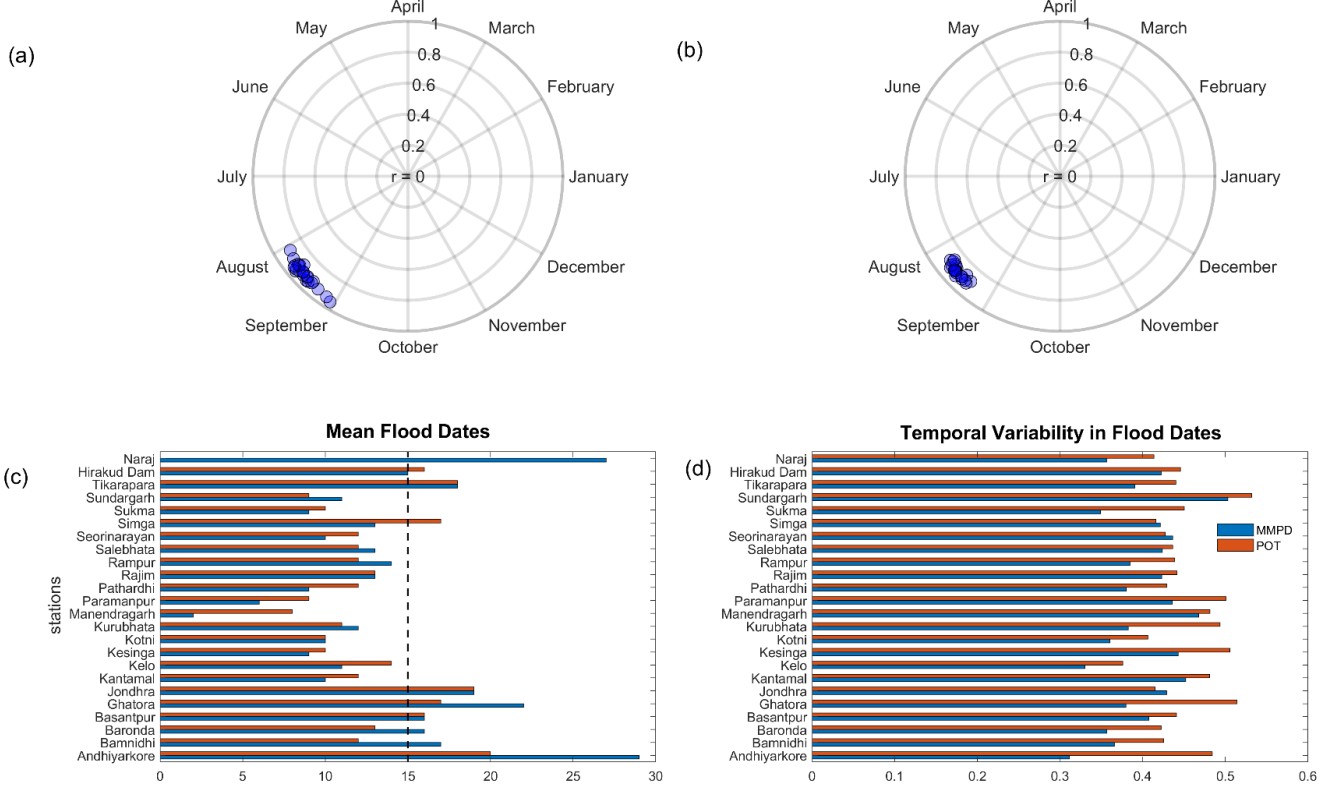

**Figure 4** Temporal distributions of mean flood dates and persistence in its timings using (a) MMF and (b) POTF time series. (c) Bar plots show the mean dates of floods. The mid of the month is marked with a black vertical line. The POTF series at Naraj gauging site contains less than a 10-year of record, hence this site is excluded from the analysis of POTF series; (d) Comparative assessments of circular variance ($\sigma^2$) in both methods of flood samplings. Figure legends apply to both subplots ($c-d$).






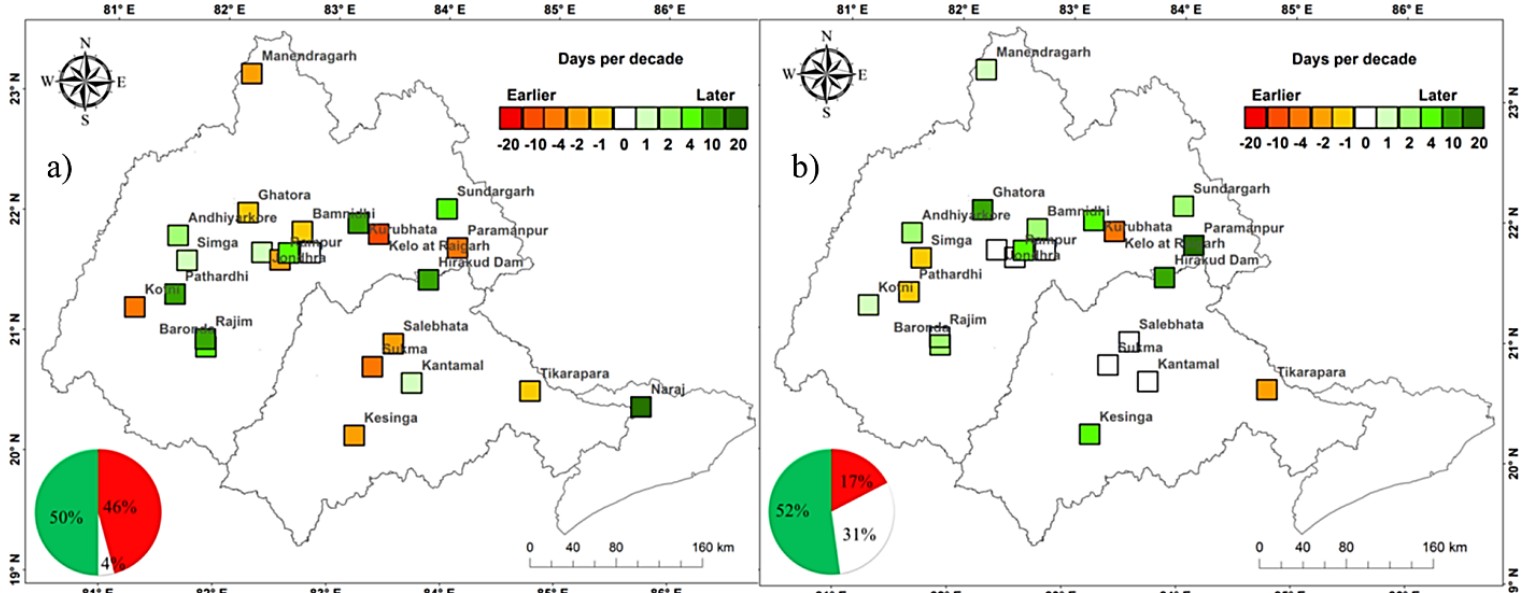

**Figure 5**. Trends in flood timing of (a) MMF and (b) POTF events at individual gauge stations. The shift in the mean date of flood occurrence is expressed in days per decade. The pie charts at the bottom left corner show the percentage of gauges with a decadal shift (*i.e.*, an earlier or delayed) in the time of flood occurrence.





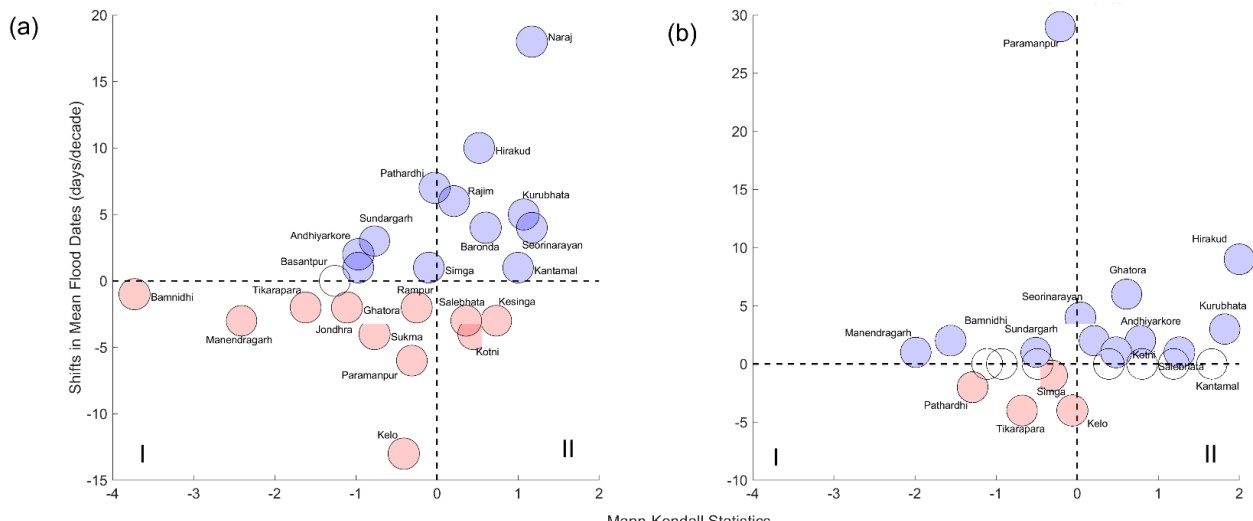

**Figure 6.** Scatter plots show sites over MRB that exhibit trends (irrespective of the level of significance) in flood magnitude and shifts in the timing of flood occurrences for (a) MMF and (b) POTF events. The sites with no shift in the timing of flood occurrences are marked in white. The sites located at quadrants I and II are the most vulnerable that show either up- or downward trends in flood magnitude coincides with an earlier date of occurrence. Sites with a delayed date of flood occurrence are marked in blue.
