# Peer review of "Co-incidence Analysis of Changes in Flood Magnitude and Shifts in Flood Timing in a Large Tropical Pluvial River Basin"

_Hydrology and Earth System Sciences, 2020_

## Referee Comment (RC1) · Anonymous Referee #1 · 4 May 2020

General comments

The manuscript "Co-incidence Analysis of Changes in Flood Magnitude and Shifts in Flood Timing in a Large Tropical Pluvial River Basin", by Ganguli et al., aims at assessing coincidence of changes in peak discharge and shifts in its timing in Mahanadi River Basin (MRB), in India.

Many of the results that have been showed for the analyzed catchment are a summary or repetition of previous studies as also the authors declare both for what concerns analysis of trends in floods (Panda et al., 2013 – lines 452-457) and trends in flood seasonality (Ganguli et al., 2020 – lines 469-473). Therefore, this study reduces to

a qualitative comparison between the two trends by introducing some explanatory hypothesis and contextualizing the finding of previous studies.

The language is fluent and precise. Scientific content and methods as well as overall presentation of the results are very good but the main problem, according to my opinion, is the poor degree of novelty of the tools, data and findings. Moreover, I strongly believe that the novel content is insufficient for the relevance of the journal. The results are presented with significant details and systematic approach. The authors made a great effort for motivating the few innovative results and study, for this reason the reading is very interesting but the paper resemblances a review paper in many paragraphs and still appears bare for what concerns new contents. Therefore, I invite authors to continue their fascinating research on the topic and to include in a future comprehensive version of the paper a supportive analysis for the new findings that can motivate the interest of the readers. For instance, I am referring to the role of catchment processes or trends in precipitation extremes for motivating the observed coincidence of trends in peak discharge and persistence in its timing.

Specific comments

I suggest changing the 3-D plot in Figure 3 in a 2-D plot to improve its readability: may be different colors and markers just help to quantify the statistics.

Line 216: typesetting error

References

Ganguli, P., Nandamuri, Y.R., and Chatterjee, C.: Analysis of Persistence in the Flood Timing and the Role of Catchment Wetness on Flood Generation in a Large River Basin in India. Theoretical and Applied Climatology, 139, 373-388, 2020.

Panda, D.K., Kumar, A., Ghosh, S., and Mohanty, R.K.: Streamflow trends in the mahanadi river basin (India): Linkages to tropical climate variability. J. Hydrol., 495, 135–149, 2013.

---

## Author Comment (AC1) · 7 May 2020

**Response to Manuscript # hess-2020-55, "Co-incidence analysis of changes in flood magnitude and shifts in flood timing in a large tropical pluvial river basin" by P. Ganguli, Y. R. Nandamuri, and C. Chatterjee**

We would like to thank the reviewer 1 for the valuable comments and for providing us an opportunity to improve our manuscript. Our responses are embedded within the comments (in BLACK) in BLUE. The new additions to the revised manuscript are embedded below in BROWN.

**Reviewer #1 (Response to Technical Comments to the Reviewer)**

**Comment 1:** The manuscript "Co-incidence Analysis of Changes in Flood Magnitude and Shifts in Flood Timing in a Large Tropical Pluvial River Basin", by Ganguli et al., aims at assessing coincidence of changes in peak discharge and shifts in its timing in Mahanadi River Basin (MRB), in India.

Many of the results that have been showed for the analyzed catchment are a summary or repetition of previous studies as also the authors declare both for what concerns analysis of trends in floods (Panda et al., 2013 – lines 452-457) and trends in flood seasonality (Ganguli et al., 2020 – lines 469-473). Therefore, this study reduces to a qualitative comparison between the two trends by introducing some explanatory hypothesis and contextualizing the finding of previous studies.

**Response:** Thanks for the feedback. Here we would like to point out to the reviewer that our study stands apart from other studies on various fronts. Taking the example of two studies that the reviewer has pointed out, for example, Panda et al. (2013) assessed trend in monthly streamflow records from 19 gauges for the period 1972 – 2007 and daily gridded rainfall records from 60 grids at 0.5° spatial resolution for the period 1972 – 2005 using Mann-Kendall trend statistics. Their analysis was limited to lower temporal resolution discharge records and establishing possible linkage between seasonal streamflow patterns and extreme rainfall indices over the Mahanadi River basin. While flow regime from higher temporal resolution would be necessary for precise estimation of flood timing and changes in magnitude and frequency of high flow, the Mann-Kendall trend statistics offer only identification of monotonic (or continuous) trends in the time series with no specific information of abrupt shift or change points within the time series. On the other hand, Ganguli et al. (2020) have showed the role of catchment wetness (considering antecedent soil moisture as a proxy variable) in modulating the timing and magnitude of floods. However, unlike previous assessments (for brevity we summarize in Table S1), that analyzed either trend in flood magnitude or the role of hydrometeorological drivers such as precipitation or antecedent soil moisture in flood generating mechanism, using high-quality daily streamflow records of 24 (out of 47 total) stream gauges, this study for the first time assesses following research questions for the Mahanadi River basin in India:

1. While other studies have explored the possibility of 'local' monotonic trends in flood record at individual river gauge locations, is the nature of trend significant at a regional level considering a collection of all sites? Second, is there any abrupt change in the flood time series and could the detected change point be linked to any major anthropogenic activity prelevant over the basin?

2. Is there any possible linkage between the trend in flood severity and catchment properties or processes, which were ignored in most of the previous assessment for a vast river network of Mahanadi?

3. Is there any evidence for the concurrence of trend (up/downward) in flood severity and shift (early/delayed occurrence) in flood timing that may help in identifying the "flood-rich and flood poor" (Merz et al., 2018) region over the basin?

**Comment 2:** The language is fluent and precise. Scientific content and methods as well as overall presentation of the results are very good but the main problem, according to my opinion, is the poor degree of novelty of the tools, data and findings. Moreover, I strongly believe that the novel content is insufficient for the relevance of the journal.

**Response:** We appreciate this comment from the reviewer. However, we do not agree with the reviewer regarding novelty aspects, quality of the data used and the scientific rigor of the study as pointed out. As discussed in the earlier response in comment 1, and also as shown in Table S1 (the last row, where we highlighted the novelty aspects of our study), we re-iterate our study is the first kind over Mahanadi River basin that investigated comprehensively the nature of floods and its shifting behavior over time using ranges of statistical tools, which were not implemented before over the study area in particular. Further, unlike earlier assessments that have used either 30 – 40 years records with lower temporal resolution flow data (for example, Panda et al., 2013 have used monthly time series of discharge) or a limited number of gauging sites (Jena et al., 2014; Mondal and Mujumdar, 2012; Panda et al., 2013), we have used the best quality up to date high-frequency daily streamflow observations of 47 years (1970 – 2016), with minimal gaps in the records from 24 gauges spatially distributed over the basins.

Our study suggests most of the sites show the earlier or delayed flood timing, which is coincided by an increased or decreased trend in flood magnitude over one-third of the gauges throughout. In summary, we find a larger number of gauges over Mahanadi basins showed a delayed shift in flood timing using both monsoonal maxima and peak-over Threshold (POT) flood series. This delayed shift in flood timing has direct implications in the operation of Hirakud dam, the longest earthen dam in India. The obtained results would provide valuable insights to inform the shifting nature of floods as a consequence of climate change and developing regional flood resilience strategies in densely populated areas of Mahanadi River basin.

**Comment 3:** The results are presented with significant details and systematic approach. The authors made a great effort for motivating the few innovative results and study, for this reason the reading is very interesting but the paper resemblances a review paper in many paragraphs and still appears bare for what concerns new contents.

**Response:** Here we sense a slight contradiction in the reviewer's comment; wherein the first part of the comment the reviewer has appreciated our work, on the other hand, s/he finds a resemblance with a review paper. Here we point out that, we have presented sufficient evidence regarding novelty aspects in the study as reflected in research questions posed, application of improved analyses methods and updated database for the study region. However, the reviewer has not clearly pointed out which part of the manuscript appears bare or which section s/he intends to be revised/deleted.

**Comment 4:** Therefore, I invite authors to continue their fascinating research on the topic and to include in a future comprehensive version of the paper a supportive analysis for the new findings that can motivate the interest of the readers. For instance, I am referring to the role of catchment processes or trends in precipitation extremes for motivating the observed coincidence of trends in peak discharge and persistence in its timing.

**Response:** We appreciate this comment from the reviewer. However, we point to the reviewer that catchment processes, such as watershed-response time or the delay between rainfall occurrences to flood events in small to medium sized catchments (as is the case for most of the catchments here) is dictated by number of factors, such as, area of watershed, flow length, topographic slope and flow resistance which is modulated by soil types (Berne et al., 2004; Gaál et al., 2012; Holtan and Overton, 1963; Kennedy and Watt, 1969). The review of the literature (Berne et al., 2004; Holtan and Overton, 1963; Kennedy and Watt, 1969) suggests, among many factors, the basin lag time can be primarily related to catchment area through a simple power law relationship; the larger (smaller) the catchment size, longer (faster) the time elapsed to propagate rain-induced runoff at downstream as a flood hydrograph. Here, we have already included, the analyses of scale-dependence to runoff sensitiveness, *i.e.*, dependence between peak discharge and catchment area in Fig. 3. Hence, our analysis has already taken care of catchment processes to runoff sensitiveness.

We further point to the reviewer that owing to the rain fed nature of the basin, a number of studies have already investigated the linkage between rainfall magnitude and extreme floods using both station-based (Panda et al., 2013; Rao, 1993, 1995) and gridded (Jena et al., 2014) rainfall records. Further, a few studies (Ghosh et al., 2010; Mondal and Mujumdar, 2012; Mujumdar and Ghosh, 2008; Raje and Mujumdar, 2009) have also investigated climate change signal over MRB using gridded precipitation records from general circulation models. Further, Ganguli et al. (2020) have

already shown the role of catchment wetness (considering antecedent soil moisture as a proxy variable) in modulating the timing and magnitude of floods.

The focus of present paper was not to reiterate the same findings but rather to investigate the possible linkage between the trend in flood severity and catchment properties or processes and identification of "flood-rich and flood poor" regions across the Mahanadi basin. Accordingly, we have removed the future research statements as explained in lines # 480 – 485 of the older version of the manuscript, which is already addressed in Ganguli et al. (2020) for Mahanadi River basin.

**Comment 5:** Specific comments

I suggest changing the 3-D plot in Figure 3 in a 2-D plot to improve its readability: may be different colors and markers just help to quantify the statistics.

**Response:** Agreed and revised.

**Comment 6:** Line 216: typesetting error.

**Response:** Agreed and revised.

**References**

Berne, A., Delrieu, G., Creutin, J.-D. and Obled, C.: Temporal and spatial resolution of rainfall measurements required for urban hydrology, Journal of Hydrology, 299(3), 166–179, doi:10.1016/j.jhydrol.2004.08.002, 2004.

Gaál, L., Szolgay, J., Kohnová, S., Parajka, J., Merz, R., Viglione, A. and Blöschl, G.: Flood timescales: Understanding the interplay of climate and catchment processes through comparative hydrology, Water Resources Research, 48(4), doi:10.1029/2011WR011509, 2012.

Ghosh, S., Raje, D. and Mujumdar, P. P.: Mahanadi streamflow: climate change impact assessment and adaptive strategies, Curr. Sci, 98(8), 1084–1091, 2010.

Holtan, H. N. and Overton, D. E.: Analyses and application of simple hydrographs, Journal of Hydrology, 1(3), 250–264, 1963.

Jena, P. P., Chatterjee, C., Pradhan, G. and Mishra, A.: Are recent frequent high floods in Mahanadi basin in eastern India due to increase in extreme rainfalls?, Journal of hydrology, 517, 847–862, 2014.

Kennedy, R. J. and Watt, W. E.: The relationship between lag time and the physical characteristics of drainage basins in Southern Ontario, 1969.

Merz, B., Dung, N. V., Apel, H., Gerlitz, L., Schröter, K., Steirou, E. and Vorogushyn, S.: Spatial coherence of flood-rich and flood-poor periods across Germany, Journal of Hydrology, 559, 813–826, 2018.

Mondal, A. and Mujumdar, P.: On the basin-scale detection and attribution of human-induced climate change in monsoon precipitation and streamflow, Water Resources Research, 48(10), 2012.

Mujumdar, P. and Ghosh, S.: Modeling GCM and scenario uncertainty using a possibilistic approach: Application to the Mahanadi River, India, Water Resources Research, 44(6), 2008.

Panda, D. K., Kumar, A., Ghosh, S. and Mohanty, R.: Streamflow trends in the Mahanadi River basin (India): Linkages to tropical climate variability, Journal of Hydrology, 495, 135–149, 2013.

Raje, D. and Mujumdar, P. P.: A conditional random field–based downscaling method for assessment of climate change impact on multisite daily precipitation in the Mahanadi basin, Water Resources Research, 45(10), 2009.

Rao, P. G.: Climatic changes and trends over a major river basin in India, Climate Research, 2, 215–223, 1993.

Rao, P. G.: Effect of climate change on streamflows in the Mahanadi River Basin, India, Water International, 20, 205–212, 1995.

---

## Referee Comment (RC2) · Anonymous Referee #2 · 18 Jun 2020

This is an important research. The paper has been written well. I have only minor comments and edits here and in the PDFs.

1. Please explicitly present the research question and hypothesis in the Introduction. 2. A schematic/flowchart of the methodology would be helpful. 3. Why the 98.5th percentile was used as the threshold level? 4. Sources of uncertainty and how they affect your results need to be discussed. 5. Please spell out all the abbreviations in the figures, tables and headings. These must stand alone.

Please also note the supplement to this comment:

[Figure]

https://www.hydrol-earth-syst-sci-discuss.net/hess-2020-55/hess-2020-55-RC2-supplement.zip

---

## Author Comment (AC2) · 20 Jun 2020

**Response to Manuscript # hess-2020-55, "Co-incidence analysis of changes in flood magnitude and shifts in flood timing in a large tropical pluvial river basin" by P. Ganguli, Y. R. Nandamuri, and C. Chatterjee**

We would like to thank the reviewer 2 for the valuable comments and for providing us an opportunity to improve our manuscript. Our responses are embedded within the comments (in BLACK) in BLUE. The new additions to the revised manuscript are embedded below in BROWN.

**Reviewer #2 (Response to Technical Comments to the Reviewer)**

**Comment 1:** This is an important research. The paper is well written. I have only minor comments and edits here and in the PDFs.

**Response:** We appreciate reviewer comments. In this revision, we have tried our best to address reviewer's comments in the subsequent paragraphs in our response letter.

**Comment 2:** Please explicitly present the research question and hypothesis in the Introduction.

**Response:** We agree. We have added following hypothesis and research questions in page 4, line # 88 of the revised manuscript:

Given challenges in flood characterizations and adaptations over Mahanadi River Basin, this paper aims to examine following hypothesis that: *the basin-wide floods are largely controlled by catchment properties, and concurrent (i.e., simultaneous) or cascading (one event preceded/succeeded by the other within a close time interval) occurrences of trends (up/downward) in flood magnitude and shifts (earlier/delayed) in its timing may further complicate the flood risks and associated impacts.*

We address the following three research questions:

1. While previous studies have explored the possibility of '*local*' monotonic trends in flood records at individual river gauge locations, is the nature of trend significant at a regional level considering a collection of all sites? In addition, is there any abrupt change in the peak discharge time series, and could the detected change point be linked to any major anthropogenic activity prevalent over the basin?

2. Is there any possible linkage between the trend in flood severity and catchment properties or processes, which were ignored in most of the previous assessment for a vast river network of Mahanadi?

3. Is there any evidence for the concurrence of trend (up/downward) in flood severity and shift (early/delayed occurrence) in flood timing that may help in identifying the "flood-rich and flood poor" (Merz et al., 2018) region over the basin?

**Comment 3:** A schematic/flowchart of the methodology would be helpful.

**Response:** Agreed. We have added the following flowchart in page 6, line # 151 of the manuscript.

[Figure]

**Figure 2. Schematics of the work flow**. The color of boxes in blue indicates input steps, yellow indicates the process steps, brown shows the expected way forward to mitigate the impact of large floods in a changing climate. While we identify monotonic trends in the flood time series using Mann-Kendall trend test considering ties and autocorrelation, the abrupt changes in the time series are detected using Pettit's change point test. We detect the persistence and shift in flood timing using circular statistics (see Methods for details). The abbreviation, *i.i.d.* indicates independent and identically distributed.

**Comment 4:** Why the 98.5$^{th}$ percentile was used as the threshold level?

**Response:** As explained in Page 6, line # 160, we evaluate different thresholds, varying from 98 to 99.9$^{th}$ percentiles at an increment of 0.5. Finally we select 98.5$^{th}$ percentile as a threshold level that allow us to choose on an average three extreme events per year. To ensure selected peak over threshold events are independent to each other, following earlier studies (Petrow and Merz, 2009; Svensson et al., 2005), we select different time spans ranging between five and ten days to decluster the partial duration time series based on the watershed area of the sub-catchments. The use of 98.5$^{th}$ percentile threshold as an indicator of peak flows from daily stream flow records and the selection of on an average three peak discharge events per year are widely used in practice to attribute extreme floods (Acero et al., 2017; Lawrence, 2020; Mangini et al., 2018; Svensson et al., 2005).

**Comment 5:** Sources of uncertainty and how they affect your results need to be discussed.

**Response:** Thanks for pointing this out. We have added following sentences in page 19, line 480 of the revised manuscript:

"While uncertainty in hydrometric observations is one of the prominent sources of uncertainty in the current analysis, especially in the data-sparse delta areas of the lower MRB, further data gathering effort would substantially enhance the confidence in the analyses. Second, spatiotemporal heterogeneity of streamflow observations remains a constraint. For instance, the uneven temporal coverage of the individual streamflow records and regional differences in the spatial distribution of gauges across three reaches of MRB is affecting the obtained results. Nonetheless, the derived insights highlights regional nature of interacting and cascading flood risks over MRB in the present-day era, which provides a stronger basis for understanding and managing such connected extremes (Raymond et al., 2020) in the future. The findings can be used to improve policy recommendations in adapting extreme floods in Anthropocene and support tools to achieve societal resilience."

**Comment 6:** Please spell out all the abbreviations in the figures, tables and headings. These must stand alone.

**Response:** Agreed and incorporated.

**References**

Acero, F. J., Parey, S., Hoang, T. T. H., Dacunha-Castelle, D., García, J. A. and Gallego, M. C.: Non-stationary future return levels for extreme rainfall over Extremadura (southwestern Iberian Peninsula), Hydrological Sciences Journal, 62(9), 1394–1411, doi:10.1080/02626667.2017.1328559, 2017.

Lawrence, D.: Uncertainty introduced by flood frequency analysis in projections for changes in flood magnitudes under a future climate in Norway, Journal of Hydrology: Regional Studies, 28, 100675, doi:10.1016/j.ejrh.2020.100675, 2020.

Mangini, W., Viglione, A., Hall, J., Hundecha, Y., Ceola, S., Montanari, A., Rogger, M., Salinas, J. L., Borzì, I. and Parajka, J.: Detection of trends in magnitude and frequency of flood peaks across Europe, Hydrological Sciences Journal, 63(4), 493–512, 2018.

Merz, B., Dung, N. V., Apel, H., Gerlitz, L., Schröter, K., Steirou, E. and Vorogushyn, S.: Spatial coherence of flood-rich and flood-poor periods across Germany, Journal of Hydrology, 559, 813–826, 2018.

Petrow, T. and Merz, B.: Trends in flood magnitude, frequency and seasonality in Germany in the period 1951–2002, Journal of Hydrology, 371(1), 129–141, doi:10.1016/j.jhydrol.2009.03.024, 2009.

Raymond, C., Horton, R. M., Zscheischler, J., Martius, O., AghaKouchak, A., Balch, J., Bowen, S. G., Camargo, S. J., Hess, J., Kornhuber, K., Oppenheimer, M., Ruane, A. C., Wahl, T. and White, K.: Understanding and managing connected extreme events, Nature Climate Change, 1–11, doi:10.1038/s41558-020-0790-4, 2020.

Svensson, C., Kundzewicz, W. Z. and Maurer, T.: Trend detection in river flow series: 2. Flood and low-flow index series/Détection de tendance dans des séries de débit fluvial: 2. Séries d'indices de crue et d'étiage, Hydrological Sciences Journal, 50(5), 2005.